# Crystal structures of human ET$_B$ receptor provide mechanistic insight into receptor activation and partial activation

Wataru Shihoya[1], Tamaki Izume[1], Asuka Inoue[2], Keitaro Yamashita [1,3], Francois Marie Ngako Kadji[2], Kunio Hirata [3], Junken Aoki[2,4], Tomohiro Nishizawa[1] & Osamu Nureki [1]

Endothelin receptors (ET$_A$ and ET$_B$) are class A GPCRs activated by vasoactive peptide endothelins, and are involved in blood pressure regulation. ET$_B$-selective signalling induces vasorelaxation, and thus selective ET$_B$ agonists are expected to be utilized for improved anti-tumour drug delivery and neuroprotection. Here, we report the crystal structures of human ET$_B$ receptor in complex with ET$_B$-selective agonist, endothelin-3 and an ET$_B$-selective endothelin analogue IRL1620. The structure of the endothelin-3-bound receptor reveals that the disruption of water-mediated interactions between W6.48 and D2.50 is critical for receptor activation, while these hydrogen-bonding interactions are partially preserved in the IRL1620-bound structure. Consistently, functional analysis reveals the partial agonistic effect of IRL1620. The current findings clarify the detailed molecular mechanism for the coupling between the orthosteric pocket and the G-protein binding, and the partial agonistic effect of IRL1620, thus paving the way for the design of improved agonistic drugs targeting ET$_B$.

[1] Department of Biological Sciences, Graduate School of Science, The University of Tokyo, Bunkyo, Tokyo 113-0033, Japan. [2] Graduate School of Pharmaceutical Sciences, Tohoku University, 6-3, Aoba, Aramaki, Aoba-ku, Sendai, Miyagi 980-8578, Japan. [3] RIKEN SPring-8 Center, Hyogo 679-5148, Japan. [4] Japan Agency for Medical Research and Development, Core Research for Evolutional Science and Technology (AMED-CREST), Tokyo 100-0004, Japan. These authors contributed equally: Wataru Shihoya, Tamaki Izume. Correspondence and requests for materials should be addressed to T.N. (email: t-2438@bs.s.u-tokyo.ac.jp) or to O.N. (email: nureki@bs.s.u-tokyo.ac.jp)

Endothelin receptors belong to the class A GPCRs, and are activated by endothelins, which are 21-amino acid peptide agonists[1]. Both of the endothelin receptors (the $ET_A$ and $ET_B$ receptors) are widely expressed in the human body, including the vascular endothelium, brain, lung, kidney, and other circulatory organs[2,3]. Three kinds of endothelins (ET-1, ET-2, and ET-3) activate the endothelin receptors (ETRs) with sub-nanomolar affinities. ET-1 and ET-2 show similar affinities to both of the endothelin receptors, while ET-3 shows two orders of magnitude lower affinity to $ET_A$[4–6]. The stimulation of the $ET_A$ receptor by ET-1 leads to potent and long-lasting vasoconstriction, whereas that of the $ET_B$ receptor induces nitric oxide-mediated vasorelaxation[7–9]. The human brain contains the highest density of endothelin receptors, with the $ET_B$ receptor comprising about 90% in areas such as the cerebral cortex[10]. The $ET_B$ receptor in neurons and astrocytes has been implicated in the promotion of neuroprotection, including neuronal survival and reduced apoptosis[11,12]. Moreover, the ET-3/$ET_B$ signalling pathway has distinct physiological roles, as compared to the ET-1 pathway. In the brain, ET-3 is responsible for salt homeostasis, by enhancing the sensitivity of the brain sodium-level sensor $Na_x$ channel[13]. The ET-3/$ET_B$ signalling pathway is also related to the development of neural crest cells, and has an essential role in the formation of the enteric nervous system[14]. Thus, mutations of the ET-3 or $ET_B$ genes cause Hirschsprung's disease, a birth defect in which nerves are missing from parts of the intestine[15,16]. Overall, the endothelin system participates in a wide range of physiological functions in the human body.

Since the activation of the $ET_B$ receptor has a vasodilating effect, unlike the $ET_A$ receptor, $ET_B$-selective agonists have been studied as vasodilator drugs for the improvement of tumour drug delivery, as well as for the treatment of hypertension[2,3]. IRL1620 (N-Suc-[E9, A11, 15] ET-1$_{8-21}$)[17], a truncated peptide analogue of ET-1, is the smallest agonist that can selectively stimulate the $ET_B$ receptor, and currently no non-peptidic $ET_B$-selective agonists have been developed. The affinity of IRL1620 to the $ET_B$ receptor is comparable to that of ET-1, whereas it essentially does not activate the $ET_A$ receptor, and thus it shows high $ET_B$ selectivity of over 100,000-fold. Due to its large molecular weight, IRL1620 is not orally active and thus requires intravenous delivery. Despite its pharmacokinetic disadvantages, IRL1620 is an attractive candidate for the treatment of various diseases related to the $ET_B$ receptor. Since the $ET_B$-selective signal improves blood flow, IRL1620 could be utilized for the improved efficacy of anti-cancer drugs by increasing the efficiency of drug delivery, as shown in rat models of prostate and breast cancer[18–21]. Moreover, this strategy can also be applied to radiotherapy in the treatment of solid tumours, as the radiation-induced reduction in the tumour volume was enhanced by IRL1620[22]. IRL1620 also has vasodilation and neuroprotection effects in the brain. IRL1620 reduced neurological damage following permanent middle cerebral artery occlusion in a rat model of focal ischaemic stroke[23]. Moreover, the stimulation of the $ET_B$ receptor by IRL1620 reduces the cognitive impairment induced by beta amyloid (1-40), a pathological hallmark of Alzheimer's disease, in rat experiments[24,25]. These data suggest that $ET_B$ selective agonists might offer new therapeutic strategies for neuroprotection and Alzheimer's disease. The safety and maximal dose of IRL1620 were investigated in a phase I study. While a recent phase 2 study of IRL1620 in combination with docetaxel as the second-line drug reported no significant improvement in the treatment of advanced biliary tract cancer (ABTC)[26], further trials for selected patients based on tumour types with various choices for the second-line drugs are still expected. Concurrently, the pharmacological properties of IRL1620 could also be improved for better clinical applications. However, little is known about the selectivity and activation

mechanism of this artificially designed agonist peptide, although the $ET_B$ structures in complex with ET-1 and antagonists have been determined[27,28].

In this study, we report the crystal structures of the $ET_B$ receptor in complex with two $ET_B$-selective ET variant agonists, ET-3 and IRL1620. Together with their detailed pharmacological characterization, the structures reveal the different activation mechanisms of these agonists, especially for the partial activation by IRL1620.

## Results

**Functional characterization of ET-3 and IRL1620.** We first investigated the biochemical activities of ET-3 and IRL1620 for the human endothelin receptors, by TGFα shedding (G-protein activation, specifically the $G_q$ and the $G_{12}$ families) and β-arrestin recruitment assays. The $EC_{50}$ and $E_{max}$ values of ET-3 for the $ET_B$ receptor were similar to those of ET-1 in both assays, while the $EC_{50}$ value for $ET_A$ was about 5-fold lower (Fig. 1a, b, and Table 1, Table 2). These data indicate that ET-3 functions as a full agonist for the endothelin receptors, with moderate $ET_B$-selectivity. The $EC_{50}$ values of IRL1620 for the $ET_B$ receptor were almost the same as those of ET-1 in both assays. In contrast, a 320 nM concentration of IRL1620 did not activate $ET_A$ in the TGFα shedding assay (Fig. 1a). These data showed that IRL1620 is $ET_B$-selective by over 3000-fold, in excellent agreement with previous functional analyses[17,29]. However, despite its sub-nanomolar affinity, the $E_{max}$ values of IRL1620 for the $ET_B$ receptor were 88% (TGFα shedding assay) and 87% (β-arrestin recruitment assay) of the $E_{max}$ value of ET-1 (Table 1 and 2), indicating that IRL1620 functions as a partial agonist for the $ET_B$ receptor.

To obtain mechanistic insights into the different actions of these agonists, we performed X-ray crystal structural analyses of the human $ET_B$ receptor in complex with ET-3 and IRL1620. For crystallization, we used the previously established, thermostabilized $ET_B$ receptor ($ET_B$-Y5)[27,30]. IRL1620 also functions as a partial agonist for the thermostabilized receptor, as the $E_{max}$ values for $ET_B$-Y5 were lower than those of ET-1 in both assays (84% and 85% in the TGFα shedding assay and the β-arrestin recruitment assay, respectively), while the $EC_{50}$ values of IRL1620 were increased for $ET_B$-Y5 by about 9- and 6-fold in the TGFα shedding assay and β-arrestin recruitment assay, respectively, as compared to the wild type receptor (Fig. 1a, b middle and Tables 1, 2). To facilitate crystallization, we replaced the third intracellular loop (ICL3) of the receptor with T4 Lysozyme ($ET_B$-Y5-T4L), and using in meso crystallization, we obtained crystals of $ET_B$-Y5-T4L in complex with ET-3 and IRL1620 (Supplementary Figure 1a, b). In total, 757 and 68 datasets were collected for the ET-3- and IRL1620-bound receptors, respectively, and merged by the data processing system KAMO[31]. Eventually, we determined the $ET_B$ structures in complex with ET-3 and IRL1620 at 2.0 and 2.7 Å resolutions, respectively, by molecular replacement using the ET-1-bound receptor (PDB 5GLH) (Table 3). The datasets for the ET-3 bound receptor were mainly collected with an automated data-collection system, ZOO, which allowed the convenient collection of a large number of datasets. The electron densities for the agonists in both structures were clearly observed in the $F_o - F_c$ omit maps (Supplementary Figure 1c, d).

**$ET_B$ structure in complex with the full agonist ET-3.** We first describe the $ET_B$ structure in complex with ET-3. The overall structure consists of the canonical 7 transmembrane helices (TM), the amphipathic helix 8 at the C-terminus (H8), two antiparallel β-strands in the extracellular loop 2 (ECL2), and the

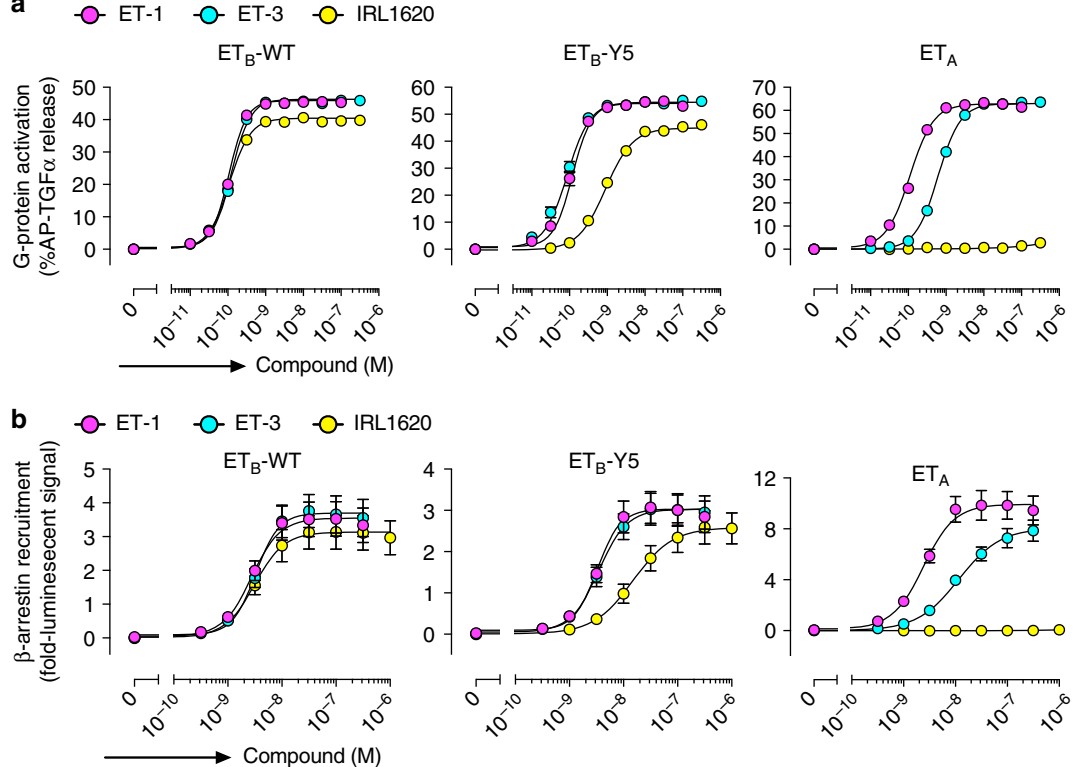

**Fig. 1** Pharmacological characterizations of ET-3 and IRL1620. **a** Concentration-response curves of AP-TGFα release in the ET-1, ET-3, and IRL1620 treatments of HEK293 cells expressing the indicated endothelin receptors. Symbols and error bars are means and s.e.m. (standard error of the mean) of five or seven independent experiments, each performed in triplicate. Note that the error bars are smaller than the symbols for most data points. **b** Concentration–response curves of β-arrestin recruitment in the ET-1, ET-3, and IRL1620 treatments of HEK293 cells expressing the endothelin receptors. Symbols and error bars are means and s.e.m. of four independent experiments, each performed in duplicate

| Table 1 TGFα shedding assay | | | | | |
|---|---|---|---|---|---|
| | ET$_B$ ($n = 7$) | | ET$_B$-Y5 ($n = 7$) | | ET$_A$ ($n = 5$) | |
| Ligand | EC$_{50}$, nM (pEC$_{50}$ ± SEM) | $E_{max}$ ± SEM, % | EC$_{50}$, nM (pEC$_{50}$ ± SEM) | $E_{max}$ ± SEM, % | EC$_{50}$, nM (pEC$_{50}$ ± SEM) | $E_{max}$ ± SEM, % |
| ET-1 | 0.11 (9.94 ± 0.04) | 100 | 0.099 (10.01 ± 0.07) | 100 | 0.13 (9.89 ± 0.04) | 100 |
| ET-3 | 0.13 (9.89 ± 0.04) | 100.5 ± 0.6 (NS) | 0.077 (10.12 ± 0.06) | 100.9 ± 0.3 (NS) | 0.65 (9.19 ± 0.03) | 101.2 ± 0.8 (NS) |
| IRL1620 | 0.11 (9.97 ± 0.03) | 87.6 ± 1.0*** | 0.92 (9.04 ± 0.04) | 83.6 ± 1.4*** | NA | NA |

The EC$_{50}$ and $E_{max}$ values of the AP-TGFα release response for the endothelin receptors. The $E_{max}$ value of the AP-TGFα release response in the ET-1 treatment was normalized to 100% for each experiment. $E_{max}$ values that significantly differ from the wild-type are denoted by asterisks
*** $P < 0.001$ as compared with ET-1, one-way ANOVA with Dunnett's post hoc test
*NS* Not significant. *NA* Parameters not available owing to lack of responses

| Table 2 β-Arrestin recruitment assay | | | | | |
|---|---|---|---|---|---|
| | ET$_B$ ($n = 4$) | | ET$_B$ ($n = 4$) | | ET$_A$ ($n = 4$) | |
| Ligand | EC$_{50}$, nM (pEC$_{50}$ ± SEM) | $E_{max}$ ± SEM, % | EC$_{50}$, nM (pEC$_{50}$ ± SEM) | $E_{max}$ ± SEM, % | EC$_{50}$, nM (pEC$_{50}$ ± SEM) | $E_{max}$ ± SEM, % |
| ET-1 | 2.7 (8.56 ± 0.02): | 100 | 3.3 (8.48 ± 0.02) | 100 | 2.3 (8.63 ± 0.05) | 100 |
| ET-3 | 3.3 (8.49 ± 0.03) | 104.8 ± 0.5 (NS) | 3.5 (8.46 ± 0.02) | 104.8 ± 0.5 (NS) | 10.8 (7.97 ± 0.05) | 81.1 ± 0.7*** |
| IRL1620 | 3.1 (8.51 ± 0.05) | 87.3 ± 2.7*** | 15.1 (7.82 ± 0.08) | 85.4 ± 3.3*** | NA | NA |

The EC$_{50}$ and $E_{max}$ values of the β-arrestin recruitment for the endothelin receptors. The $E_{max}$ value of the β-arrestin recruitment in the ET-1 treatment was normalized to 100% for each experiment. $E_{max}$ values that significantly differ from the wild-type are denoted by asterisks
***$P < 0.001$ as compared with ET-1, one-way ANOVA with Dunnett's post hoc test
*NS* Not significant. *NA* Parameters not available owing to lack of responses

**Table 3 Data collection and refinement statistics**

|  | ET-3 | IRL1620 |
|---|---|---|
| **Data collection** | | |
| Space group | $C222_1$ | $C222_1$ |
| Cell dimensions | | |
| $a, b, c$ (Å) | 65.5, 172.3, 121.3 | 100.0, 303.9, 60.2 |
| $\alpha, \beta, \gamma$ (°) | 90, 90, 90 | 90, 90, 90 |
| Resolution (Å)[a] | 50-2.00 (2.12-2.00) | 50-2.70 (2.80-2.70) |
| $R_{meas}$[a] | 0.860 (18.057) | 0.499 (5.826) |
| $R_{pim}$[a] | 0.060 (1.277) | 0.108 (1.250) |
| $<I/\sigma(I)>$[a] | 20.7 (0.95) | 9.0 (1.4) |
| $CC_{1/2}$[a] | 0.998 (0.559) | 0.981 (0.382) |
| Completeness (%)[a] | 100.0 (100.0) | 99.99 (100.0) |
| Redundancy[a] | 204.6 (199.2) | 21.5 (21.5) |
| **Refinement** | | |
| Resolution (Å) | 50-2.00 | 50-2.70 |
| No. of reflections | 46,785 | 25,764 |
| $R_{work}/R_{free}$ | 0.1834/0.2271 | 0.2066/0.2335 |
| No. of atoms | | |
| Protein | 3914 | 3871 |
| Ligand/ion | 296 | 108 |
| Water | 175 | 59 |
| $B$-factors ($Å^2$) | | |
| Protein | 44.5 | 64.8 |
| Ligand/ion | 79.2 | 80.8 |
| Water | 49.7 | 40.8 |
| R.m.s. deviations from ideal | | |
| Bond lengths (Å) | 0.007 | 0.002 |
| Bond angles (°) | 0.862 | 0.517 |
| Ramachandran plot | | |
| Favoured (%) | 98.6 | 97.3 |
| Allowed (%) | 1.2 | 2.5 |
| Outlier (%) | 0.2 | 0.2 |

[a]Values in parentheses are for highest-resolution shell

N-terminus that is anchored to TM7 by a disulfide bond (Fig. 2b), and is similar to the previous ET-1-bound structure[27] (overall R. M.S.D of 1.0 Å for the $C_\alpha$ atoms) (Fig. 2c). Similar to ET-1, ET-3 adopts a bicyclic architecture comprising the N-terminal region (residues 1–7), the α-helical region (residues 8–17), and the C-terminal region (residues 18–21), and the N-terminal region is attached to the central α-helical region by the intrachain disulfide bond pairs (C1–C15 and C3–C11). The amino acid residues of the α-helical and C-terminal regions are highly conserved between ET-1 and ET-3 (Fig. 2a), and the agonist peptides superimposed well (Fig. 2c, d and Supplementary Figure 2a–d). Accordingly, these regions form similar interactions with the receptor in both structures (Supplementary Figure 3a, b). In contrast, all of the residues, except for the disulfide bond-forming C1 and C3, are replaced with bulkier residues in ET-3 (Fig. 2a). Despite these sequence differences, the N-terminal regions are similarly accommodated in the orthosteric pocket in both structures, because these bulky residues are exposed to the solvent and interact poorly with the receptor (Fig. 2d). These structural features explain the similar high affinity binding of ET-3 to the $ET_B$ receptor, as compared with ET-1.

Previous studies demonstrated that the N-terminal residues of the $ET_B$ receptor have a critical role in the virtually irreversible binding of the endothelins[32]. As in the ET-1-bound structure, the N-terminal tail is anchored to TM7 via a disulfide bond between C90 and C358 in the ET-3-bound structure, constituting a lid that prevents agonist dissociation. The high-resolution ET-3-bound structure allowed more accurate tracing of the elongated N-terminal residues (Fig. 2e, f, and Supplementary Figure 1e), as

compared with the ET-1-bound structure, and revealed more extensive interactions with the agonist peptide. P88, I94, Y247[ECL2], and K248[ECL2] form a lid over ET-3, which is stabilized by a water-mediated hydrogen bonding network among the carbonyl oxygen of P93, the side chains of Y247[ECL2] and K248[ECL2], and D8 of ET-3 (Fig. 2e). In addition, three consecutive prolines (P87, 88, 89) stretch over the N-terminal region of ET-3, and two of them form van der Waals interactions with ET-3. Moreover, ECL1, 2 and the N-terminal residues form an extended water-mediated hydrogen bonding network over ET-3. These extensive interactions strongly prevent the agonist dissociation.

**$ET_B$ structure in complex with the partial agonist IRL1620.** Next we describe the $ET_B$ structure in complex with the partial agonist IRL1620, a linear peptide analogue of ET-1[17] (Fig. 3a). Previous mutant and structural studies revealed that the N-terminal region contributes to the stability of the overall bicyclic structure by the intramolecular disulfide bonds, and thus facilitates the receptor interaction[17,27,33]. IRL1620 completely lacks the N-terminal region, and consists of only the α-helical and C-terminal regions (Fig. 3b). Two cysteines in the α-helical region are replaced with alanines, and negative charges are introduced into the N-terminal end of the helix, by replacing lysine with glutamic acid (E9) and modifying the N-terminal amide group with a succinyl group (Fig. 3c). The consequent cluster of negative charges on the N-terminal end of IRL1620 (succinyl group, D8, E9, and E10) reportedly has an essential role in IRL1620 binding to the $ET_B$ receptor[17]. This cluster electrostatically complements the positively charged $ET_B$ receptor pocket, which includes K346[6.58] and R357[ECL3] (Fig. 3d). Moreover, this negative cluster probably stabilizes the α-helical conformation of IRL1620, by forming a hydrogen-bonding cap at the N-terminally exposed amines[34]. Due to these effects, IRL1620 adopts a similar helical conformation, even without the intramolecular disulfide bonds (Supplementary Figure 2c, d), and forms essentially similar interactions with the receptor, as compared with the endogenous agonists, ET-1 and ET-3 (Supplementary Figure 3a–c). These structural features are consistent with the high affinity of IRL1620 to the $ET_B$ receptor, which is comparable to those of ET-1 and ET-3 (Fig. 1a, b). Nevertheless, the N-terminal side of the α-helical region of IRL1620 is less visualized in the electron density, suggesting its higher flexibility as compared to those of ET-1 and ET-3 (Supplementary Figure 1d), probably due to the lack of the N-terminal region. In contrast to the quasi-irreversible binding of ET-1, IRL1620 binding is reportedly reversible[35]. Such structural differences may account for the different dissociation properties of these agonists.

IRL1620 does not bind to the $ET_A$ receptor at all in the same concentration range, confirming its high selectivity for the $ET_B$ receptor (Fig. 1a, b, and Table 1 and 2). To elucidate the mechanism of this selectivity, we compared the amino acid compositions of the IRL1620 binding sites between the $ET_B$ and $ET_A$ receptors (Fig. 4a and Supplementary Figure 4). While the transmembrane region is highly conserved, the residues in ECL2 are diverse. In particular, the hydrophobic residues L252[ECL2] and I254[ECL2] are replaced with the polar residues H236 and T238 in the $ET_A$ receptor, respectively (Fig. 4a, b). These residues form extensive hydrophobic interactions with the middle part of IRL1620. However, the double mutation of L252H and I254T only reduced the potency of IRL1620 by 2-fold in the TGFα shedding assay (Fig. 4c and Table 4), suggesting that these residues are not the sole determinants for the receptor selectivity of IRL1620. Therefore, we focused on other residues of ECL2. In the $ET_B$ receptor, P259[ECL2] and V260[ECL2] generate a short kink

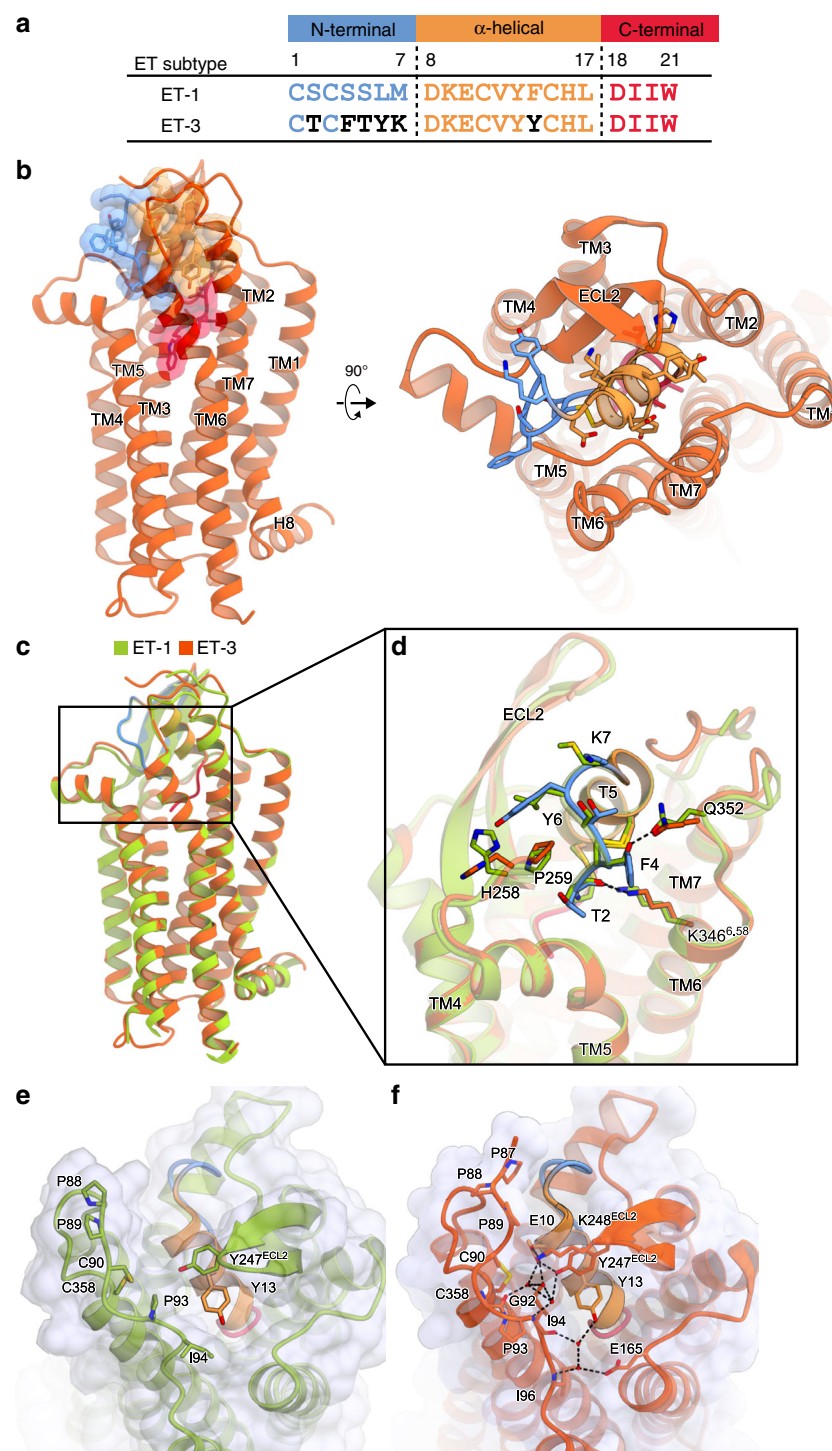

**Fig. 2** ET$_B$ structure in complex with ET-3. **a** Comparison of the amino acid sequences of ET-1 and ET-3. **b** The overall structure of the ET-3-bound ET$_B$ receptor. The receptor is shown as an orange-red ribbon model. ET-3 is shown as a transparent surface representation and a ribbon model, with its N-terminal region coloured cyan, α-helical region orange, and C-terminal region deep pink. The side chains of ET-3 are shown as sticks. **c, d** Superimposition of the ET-3 and ET-1-bound ET$_B$ receptors, coloured orange-red and green, respectively, viewed from the membrane plane (**c**), and from the extracellular side (**d**). The side chains of ET-1 and ET-3 are shown as sticks. **e, f** Comparison of the interactions in the ET-1- (**e**) and ET-3- (**f**) bound structures. The receptors are shown as transparent surface representations and ribbon models. Endothelins are shown as ribbon models. The residues involved in the irreversible binding of the endothelins are shown as sticks

on the loop between the β-strand and TM5, but the ET$_A$ receptor has a truncated loop region and completely lacks these residues (Fig. 4a). In addition, the ET$_A$ receptor has a proline (P228) in the first half of ECL2, which should disturb the β-strand formation as

in the ET$_B$ receptor (Fig. 4b). These observations suggest that ECL2 adopts completely different structures between the ET$_A$ and ET$_B$ receptors. Moreover, ECL1 is also different between the two receptors, as the ET$_A$ receptor has a five amino-acid elongation as

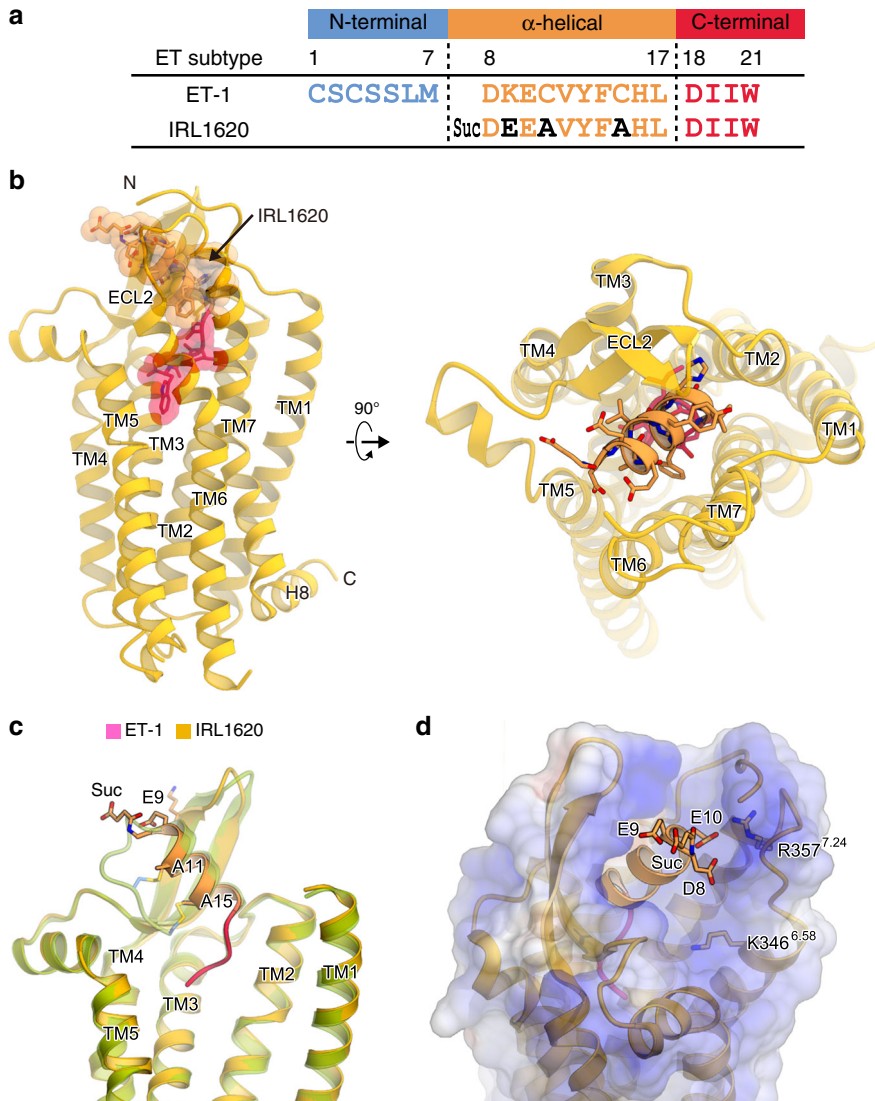

**Fig. 3** ETB structure in complex with IRL1620. **a** Comparison of the amino acid sequences of ET-1 and IRL1620. **b** The overall structure of the IRL1620-bound ETB receptor. The receptor is shown as an orange ribbon model. IRL1620 is shown as a transparent surface representation and a ribbon model, with the α-helical and C-terminal regions coloured orange and deep pink, respectively. **c** Superimposition of the IRL1620- and ET-1-bound receptors viewed from the membrane plane, coloured orange and green, respectively. The ETB receptors and agonists are shown as ribbon models. The different residues between ET-1 and IRL1620 are shown as sticks. **d** Electrostatic surfaces of the IRL1620-bound ETB structure. The negatively charged moieties on the N-terminal end of IRL1620 and the positively charged residues in the extracellular side of TMs 6 and 7 are shown as sticks

compared to the ETB receptor (Supplementary Figure 4). Since ECL1 interacts with the β-strands in ECL2 (Fig. 4a), this elongation could affect the orientation of the β-strands. Overall, the sequence divergences in the extracellular loops suggest that the ETA and ETB receptors have different extracellular architectures in these regions, which may account for their different selectivities for the isopeptides.

**Receptor activation and partial activation.** To elucidate the mechanism of the partial activation by IRL1620, we compared the IRL1620-bound structure with the full-agonist ET-3-bound structure (Fig. 5). IRL1620 forms essentially similar receptor interactions to those of the α-helical and C-terminal regions of ET-3 (Supplementary Figure 5a, b). The intracellular portions of the receptors are quite similar between the ET-3- and IRL1620-bound structures, in which TM7 and H8 adopt active conformations, while the remaining parts of the receptors still represent the inactive conformation of GPCRs (Fig. 5a, and Supplementary

Figure 6). On the extracellular side, IRL1620 induces similar conformational changes to those observed in the ET-1- and ET-3-bound structures; namely, the large inward motions of TM2, 6, and 7, which are critical for receptor activation (Fig. 5b, c). However, the extent of the inward motion of TM6-7 is smaller by about 1 Å in the IRL1620-bound structure, as compared with that in the ET-3-bound structure, due to the different ligand architectures between IRL1620 and ET-3. Since IRL1620 lacks the N-terminal region, the orthosteric pocket of the receptor has more space, and consequently the α-helical region of IRL1620 is tilted differently toward TM6 (Fig. 5b). In addition, while the N-terminal region of ET-3 interacts with TM6 of the receptor, by forming a hydrogen bond between the carbonyl oxygen of T2 and K346$^{6.58}$ (superscripts indicate Ballesteros–Weinstein numbers), IRL1620 lacks this interaction, resulting in the different orientation of TM6-7. As TM6 has an especially important role for the cytoplasmic G-protein binding, this difference is probably related to the partial agonist activity of IRL1620.

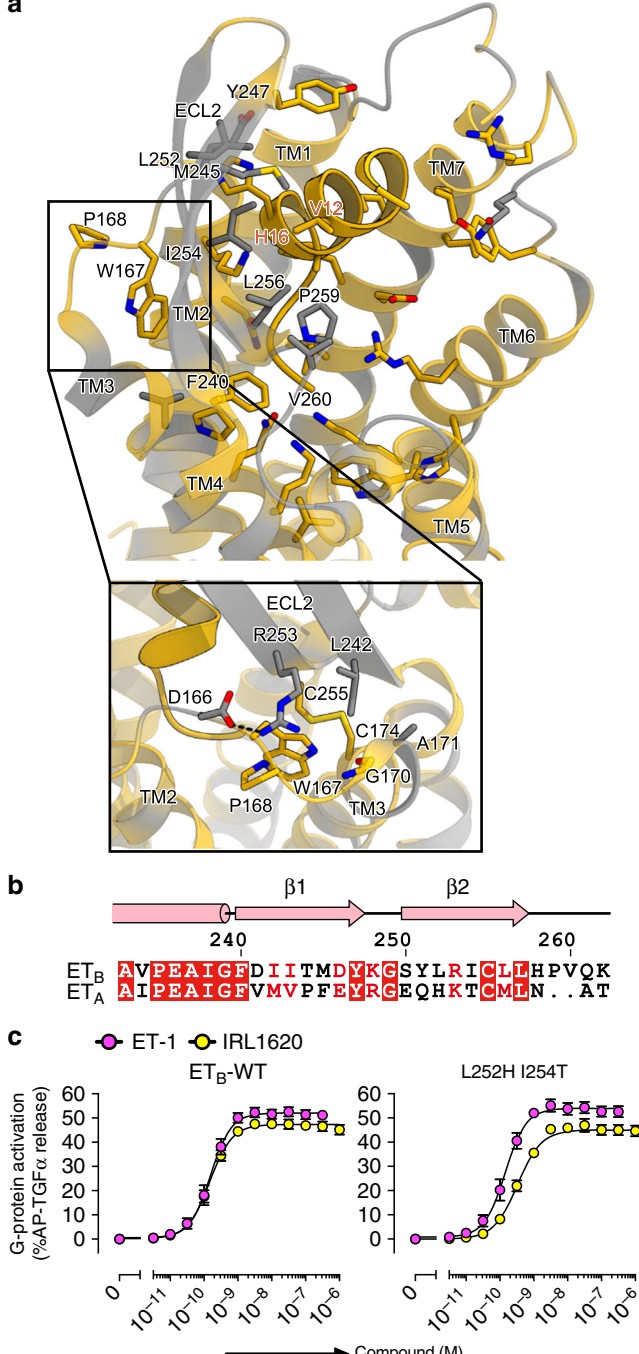

**Fig. 4** Conservation of the IRL1620 binding site. **a** Sequence conservation between ET$_A$ and ET$_B$, mapped onto the IRL1620-bound structure. The upper panel shows the conservation of the residues constituting the IRL1620 binding site. Conserved and non-conserved residues are coloured orange and grey, respectively. The ET$_B$ receptor is shown as ribbons, and the residues involved in IRL1620 binding are shown as sticks. Moreover, W167 and P168 in ECL1 are shown as sticks. The lower panel shows the conservation of the ECL1-ECL2 interface. **b** Alignment of the amino acid sequences of the human ET$_B$ and ET$_A$ receptors, focused on ECL2. **c** Concentration-response curves of AP-TGFα release response upon ET-1 or IRL1620 treatments of HEK293 cells expressing the wild-type (WT) ET$_B$ receptor or the L252H I254T double mutant. Symbols and error bars are means and s.e.m. of six independent experiments, each performed in triplicate

### Table 4 Pharmacological parameters for the L252H I254T double mutant

|  | ET$_B$-WT ($n = 6$) | L252H/I254T ($n = 6$) |
| --- | --- | --- |
| Ligand | EC$_{50}$, nM (pEC$_{50}$ ± SEM) | EC$_{50}$, nM (pEC$_{50}$ ± SEM) |
| ET-1 | 0.15 (9.82 ± 0.11) | 0.14 (9.85 ± 0.11, NS) |
| IRL1620 | 0.14 (9.85 ± 0.09) | 0.32 (9.50 ± 0.07*) |

pEC$_{50}$ value that significantly differs from the wild-type is denoted by an asterisk
*$P < 0.05$ as compared with ET$_B$-WT, Sidak's multiple comparisons test
NS Not significant

A comparison of the intermembrane parts revealed further differences in the allosteric coupling between the orthosteric pocket and the intermembrane part. Previous studies have shown that the agonist binding induces the disruption of the hydrogen-bonding network around D147$^{2.50}$, which connects TMs 2, 3, 6, and 7 and stabilizes the inactive conformation of the ET$_B$ receptor[28] (Supplementary Figure 7a). The present high-resolution ET-3-bound structure provides a precise mechanistic understanding of this rearrangement (Fig. 5d, and Supplementary Figure 7b). In particular, the water-mediated hydrogen bonds involving D147$^{2.50}$, W336$^{6.48}$, and N378$^{7.45}$ in the inactive conformation collapse upon ET-3 binding, by the inward motions of TMs 2, 6, and 7. The W336$^{6.48}$ side chain moves downward by about 2.5 Å, resulting in the disruption of the water-mediated hydrogen bond with D147$^{2.50}$, and consequently, the D147$^{2.50}$ side chain moves downward by about 3 Å and forms hydrogen bonds with the N382$^{7.49}$ and N119$^{1.50}$ side chains. The N378$^{7.45}$ side chain also moves downward by about 1.5 Å and forms a hydrogen bond with the nitrogen atom of the W336$^{6.48}$ side chain. The downward movements of the W336$^{6.48}$ and N378$^{7.45}$ side chains consequently induce the outward repositioning of the F332$^{6.44}$ side chain and the middle part of TM6, by about 1 Å. W6.48 and F6.44 are considered to be the transmission switch of the class A GPCRs, which transmits the agonist-induced motions to the cytoplasmic G-protein coupling interface. Overall, our results show that the collapse of the water-mediated hydrogen-bonding network involving D147$^{2.50}$, W336$^{6.48}$, and N378$^{7.45}$ propagates as the structural change in the transmission switch, and probably induces the outward displacement of the cytoplasmic portion of TM6 upon G-protein activation (Supplementary Figure 6).

IRL1620 induces a similar but slightly different rearrangement of the hydrogen bonding network in the intermembrane part (Fig. 5e). Due to the smaller inward shift of the extracellular portion of TM6, the downward shift of the W336$^{6.48}$ side chain is smaller in the IRL1620-bound structure, and it still forms a water-mediated hydrogen bond with the D147$^{2.50}$ side chain. Consequently, the D147$^{2.50}$ side chain forms a direct hydrogen bond with N378$^{7.45}$, thereby preventing the downward motion of N378$^{7.45}$ and the hydrogen bond formation between N378$^{7.45}$ and W336$^{6.48}$. Overall, the downward motions of the W336$^{6.48}$ and N378$^{7.45}$ side chains are only moderate, as compared to those in the ET-3-bound structure, and the hydrogen-bonding network involving D147$^{2.50}$, W336$^{6.48}$, and N378$^{7.45}$ is partially preserved in the IRL1620-bound structure (Fig. 5e, and Supplementary Figure 7c). Accordingly, in the IRL1620-bound structure, the position of the "transmission switch" residue F332$^{6.44}$ is in between those of the active (ET-3-bound) and inactive (K-8794-bound) structures (Fig. 5f). This intermediate position of F332$^{6.44}$ should partly prevent the outward displacement of the cytoplasmic portion of TM6 that is required for G-protein

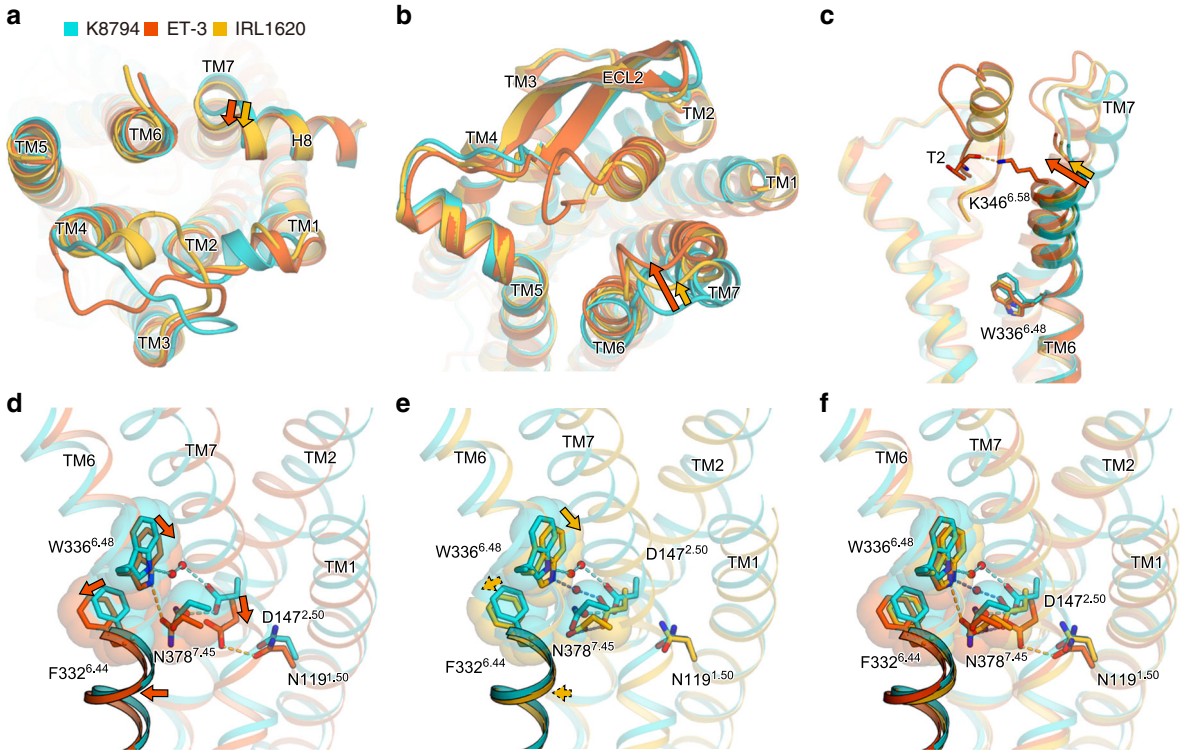

**Fig. 5** Comparison of the K8794, ET-3, and IRL1620-bound structures. **a–c** Comparison of the K-8794-bound inactive (PDB 5X93), IRL1620-bound partially active, and ET-3-bound active ET$_B$ structures, coloured turquoise, orange, and orange-red, respectively. The ET$_B$ receptors and agonist peptides are shown as ribbon models, viewed from the intracellular side (**a**), the extracellular side (**b**), and the membrane plane (**c**). The cytoplasmic cavity for the G-protein binding is still hindered by the inwardly-oriented TM6 in the ET-3 and IRL1620-bound receptors, representing the inactive conformations. This is consistent with the notion that the fully active conformation is only stabilized when the G-protein is bound, as shown in the previous nuclear magnetic resonance (NMR) and double electron–electron resonance (DEER) spectroscopy study[38]. **d–f** Superimposition of the ET$_B$ structures bound to K-8794 and ET-3 (**d**), K-8794 and IRL1620 (**e**), and K-8794, IRL1620, and ET-3 (**f**), focused on the intermembrane parts. The receptors are shown as ribbons, and the side chains of N119[1.50], D147[2.50], F332[6.44], W336[6.48], and N378[7.45] are shown as sticks with transparent surface representations. Waters are shown as red spheres. The dashed lines show hydrogen bonds coloured in the respective structures. The arrows show the structural changes on ligand binding

activation. Overall, the smaller inward shift of the extracellular potion of TM6 and the preserved interactions at the receptor core account for the partial agonistic activity of IRL1620 (Fig. 6).

## Discussion

Previous studies have suggested that the α-helical and C-terminal regions of endothelins are critical elements for receptor activation[33,36,37], whereas the N-terminal region is only responsible for the ETR selectivity[6]. Indeed, the N-terminal region-truncated analogue IRL1620 has similar EC$_{50}$ values, as compared with ET-1[17]. However, our pharmacological experiments for the first time proved that IRL1620 functions as a partial agonist for the ET$_B$ receptor, rather than a full agonist, suggesting the participation of the N-terminal region in the activation process of the ET$_B$ receptor. To clarify the receptor activation mechanism, we determined the crystal structures of the human ET$_B$ receptor in complex with ET-3 and IRL1620. The high-resolution structure of the ET-3-bound ET$_B$ receptor revealed that the large inward motions of the extracellular portions of TMs 2, 6, and 7 disrupt the water-mediated hydrogen bonding network at the receptor core (Fig. 5d, and Supplementary Figure 7a, b). The IRL1620-bound ET$_B$ structure revealed that the IRL1620-induced inward motions of TMs 6 and 7 are smaller by about 1 Å, as compared with those caused by ET-3 (Fig. 5b, c). Despite the lower resolution of the IRL1620-bound structure, the $2F_o - F_c$ map shows different rearrangement of water molecules and amino acid

residues in the receptor core, in which the hydrogen-bonding network is partially preserved (Fig. 5e, and Supplementary Figure 7c). This preserved network, together with the smaller inward motion of TM6, may prevent cytoplasmic outward motion of TM6 that occurs upon G-protein binding. These observations suggest that the interactions between the N-terminal regions of endothelins and TM6 also participate in receptor activation, while the extensive interactions of the α-helical and C-terminal regions with the receptor primarily contribute to this process (Supplementary Figure 3c). This activation mechanism is different from that of the small-molecule activated GPCRs (e.g., β2 adrenaline and M2 muscarinic acetylcholine receptors), in which only a small number of hydrogen-bonding interactions between the agonist and the receptor induce receptor activation, by affecting the receptor dynamics[38,39].

D2.50 is one of the most conserved residues among the class A GPCRs (90%). Recent high-resolution structures have revealed that a sodium ion coordinates with D2.50 and forms a water-mediated hydrogen bonding network in the intermembrane region, which stabilizes the inactive conformation of the receptor[40], and its collapse leads to receptor activation. Our previous 2.2 Å resolution structure of the K-8794-bound ET$_B$ receptor revealed that a water molecule occupies this allosteric sodium site and participates in the extensive hydrogen-bonding network, instead of a sodium ion[28] (Supplementary Figure 7a), and this hydrogen-bonding network is collapsed in the 2.8 Å resolution structure of the ET-1-bound ET$_B$ receptor, indicating its involvement in the receptor activation[27]. Nevertheless, the precise

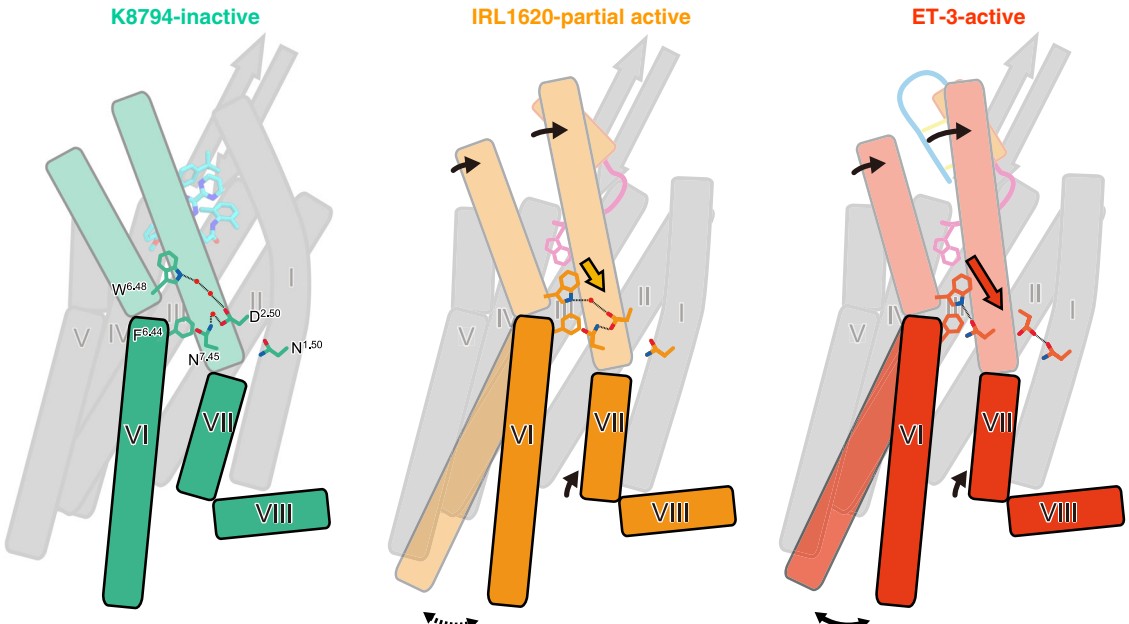

**Fig. 6** Receptor activation by ET-3 and partial activation by IRL1620. Schematic representations of receptor activation by ET-3 and partial activation by IRL1620. TM6, TM7, and H8 are highlighted. The residues involved in the signal transduction (N1.50, D2.50, F6.44, W6.48, and N7.45) are represented with sticks. Hydrogen bonds are indicated by black dashed lines. Arrows indicate the conformational changes in TM6 and TM7 upon ET-3 and IRL1620 binding. In the K-8794-bound structure (left), a water-mediated hydrogen bonding network among D147[2.50], W336[6.48], and N378[7.45] stabilizes the inactive conformation of the receptor. ET-3 binding disrupts this network and propagates the structural change of the transmission switch comprising F332[6.44] and W336[6.48] (middle), leading to the outward movement of TM6 upon G-protein coupling. In the IRL1620-bound structure (right), the water-mediated hydrogen-binding network is preserved, thus preventing the outward movement of TM6 upon G-protein coupling

rearrangement of this network still remained to be elucidated, due to the limited resolution. The current 2.0 Å resolution structure of the ET-3-bound $ET_B$ receptor revealed that the collapse of the water-mediated interaction between W336[6.48] and D147[2.50] is critical for receptor activation (Fig. 5d). This network is still partly preserved in the IRL1620-bound structure (Fig. 5e), thus preventing the transition to the fully active conformation upon G-protein coupling (Fig. 6). W6.48 is also highly conserved among the class A GPCRs (71%)[41], and the association between W6.48 and D2.50 has a critical role in the GPCR activation process, as shown in the previous nuclear magnetic resonance (NMR) study of the adenosine $A_{2A}$ receptor[42]. Given the importance of W3.36 and D2.50 in the activation of GPCRs, our proposed model of the partial receptor activation by IRL1620 is consistent with the previous functional analyses of GPCRs. To date, the $\beta_1$ adrenergic receptor is the only receptor for which agonist- and partial agonist-bound structures were reported[43]. However, these structures are both stabilized in inactive conformations by the thermostabilizing mutations and thus revealed only slight differences (Supplementary Figure 8). Therefore, our study provides the first structural insights into the partial activation of class A GPCRs.

Our current study further suggests possible improvements in clinical studies using $ET_B$-selective agonists. IRL1620 is the smallest among the $ET_B$-selective agonists, and thus is expected to be useful for the treatment of cancers and other diseases[18–22,24,25]. While its effectiveness has been proven in rat experiments, a recent phase 2 study has failed[26], and thus further improvement of IRL1620 is required for clinical applications. Our cell-based assays and structural analysis revealed the partial agonistic effect of IRL1620 on the $ET_B$ receptor in the G-protein coupling and β-arrestin recruitment activities, suggesting the possible tuning of its efficacy. The development of $ET_B$-selective agonists by fine-tuning their G-protein activation and/or β-

arrestin recruitment activities might be beneficial for clinical applications.

## Methods

**Expression and purification.** We used the thermostabilized receptor $ET_B$-Y5-T4L. as previously established (Supplementary Table 1). In brief, the haemagglutinin signal peptide followed by the Flag epitope tag was added to the N-terminus, and a tobacco etch virus (TEV) protease recognition sequence was introduced between G57 and L66. The C-terminus was truncated, and three cysteine residues were mutated to alanine (C396A, C400A, and C405A). To facilitate crystallogenesis, T4 lysozyme (C54T and C97A) was introduced into intracellular loop 3.

The $ET_B$-Y5-T4L was subcloned into a modified pFastBac vector, with the resulting construct encoding a TEV cleavage site followed by a GFP-His[10] tag at the C-terminus. The recombinant baculovirus was prepared using the Bac-to-Bac baculovirus expression system (Invitrogen). Sf9 insect cells (Invitrogen) were infected with the virus at a cell density of $4.0 \times 10^6$ cells per millilitre in Sf900 II medium, and grown for 48 h at 27 °C. The cells were disrupted by sonication, in buffer containing 20 mM Tris–HCl, pH 7.5, and 20% glycerol. The membrane fraction was collected by ultracentrifugation, and solubilized in buffer, containing 20 mM Tris–HCl, pH 7.5, 200 mM NaCl, 1% DDM, 0.2% cholesterol hemisuccinate, and 2 mg/ml iodoacetamide, for 1 h at 4 °C. The insoluble material was removed by ultracentrifugation at $180,000 \times g$ for 20 min, and incubated with TALON resin (Clontech) for 30 min. The resin was washed with ten column volumes of buffer, containing 20 mM Tris–HCl, pH 7.5, 500 mM NaCl, 0.1% LMNG, 0.01% CHS, and 15 mM imidazole. The receptor was eluted in buffer, containing 20 mM Tris–HCl, pH 7.5, 500 mM NaCl, 0.01% LMNG, 0.001% CHS, and 200 mM imidazole. TEV protease was added to the eluate, and the mixture was dialysed against buffer (20 mM Tris–HCl, pH 7.5, and 500 mM NaCl). The cleaved GFP and the protease were removed with $Co^{2+}$-NTA resin. The flowthrough was concentrated and loaded onto a Superdex200 10/300 Increase size-exclusion column, equilibrated in buffer containing 20 mM Tris–HCl, pH 7.5, 150 mM NaCl, 0.01% LMNG, and 0.001% CHS. Peak fractions were concentrated to 40 mg ml$^{-1}$ using a centrifugal filter device (Millipore 50 kDa MW cutoff), and frozen until crystallization. ET-3 or IRL1620 was added to a final concentration of 100 μM, during the concentration.

**Crystallization.** The purified receptors were reconstituted into molten lipid (monoolein and cholesterol 10:1 by mass) at a weight ratio of 1:1.5 (protein:lipid). The protein-laden mesophase was dispensed into 96-well glass plates in 30 nl drops and overlaid with 800 nl precipitant solution, using an LCP dispensing robot

(Gryphon, Art Robbins Instruments)[44,45]. Crystals of $ET_B$-Y5-T4L bound to ET-3 were grown at 20 °C in the precipitant conditions containing 25% PEG500DME, 100 mM MES-NaOH, pH 6.0, and 100 mM ammonium citrate tribasic. The crystals of $ET_B$-Y5-T4L bound to IRL1620 were grown in the precipitant conditions containing 20–25% PEG500DME, 100 mM sodium citrate, pH 5.0, and 100 mM $(NH_4)_2SO_4$ or $NH_4Cl$. The crystals were harvested directly from the LCP using micromounts (MiTeGen) or LithoLoops (Protein Wave) and frozen in liquid nitrogen, without adding any extra cryoprotectant.

**Data collection and structure determination**. X-ray diffraction data were collected at the SPring-8 beamline BL32XU with $1 \times 10$ to $8 \times 25 \, \mu m^2$ (width × height) micro-focused beams and an EIGER X 9M detector (Dectris). For the IRL1620 data, we manually collected 68 data sets (10°–180° per crystal), and the collected images were automatically processed with KAMO[31] (https://github.com/keitaroyam/yamtbx). Each data set was indexed and integrated with XDS[46] and then subjected to a hierarchical clustering analysis based on the unit cell parameters using BLEND[47]. After the rejection of outliers, 46 data sets were finally merged with XSCALE[46]. From the ET-3-bound crystals, various wedge datasets (3–180°) per crystal were mainly collected with the ZOO system, an automatic data-collection system developed at SPring-8. The loop-harvested microcrystals were identified by raster scanning and subsequently analysed by SHIKA[48]. The collected images were processed in the same manner, except that correlation coefficient-based clustering was used instead of BLEND, and finally 483 datasets were merged. The ET-3-bound structure was determined by molecular replacement with PHA-SER[49], using the ET-1-bound $ET_B$ structure (PDB 5GLH). Subsequently, the model was rebuilt and refined using COOT[50] and PHENIX[51], respectively. The IRL1620-bound structure was determined by molecular replacement, using the ET-1-bound $ET_B$ structure, and subsequently rebuilt and refined as described above. The final model of ET-3-bound $ET_B$-Y5-T4L contained residues 86–303 and 311–403 of $ET_B$, all residues of T4L, ET-3, 12 monoolein molecules, a citric acid, and 175 water molecules. The final model of IRL1620-bound $ET_B$-Y5-T4L contained residues 87–303 and 311–402 of $ET_B$, all residues of T4L, IRL1620, 3 monoolein molecules, 4 sulfate ions, a citric acid, and 59 water molecules. The model quality was assessed by MolProbity[52]. Figures were prepared using CueMol (http://www.cuemol.org/en/)

**TGFα shedding assay**. The TGFα shedding assay[53], which measures the activation of $G_q$ and $G_{12}$ signalling, was performed as described previously[28]. Briefly, a pCAG plasmid encoding a codon-optimized $ET_B$ construct (Supplementary Table 2) with an internal FLAG epitope tag or a full-length human $ET_A$ construct with an internal FLAG epitope tag cloned from cDNA was prepared. Double mutant L252H I254T was generated by Quickchange PCR (Supplementary Table 3). These plasmids were transfected, together with a plasmid encoding alkaline phosphatase (AP)-tagged TGFα (AP-TGFα), into HEK293 cells by using a polyethylenimine (PEI) transfection reagent (1 μg ETR plasmid, 2.5 μg AP-TGFα plasmid and 25 μl of 1 mg/ml PEI solution per 10-cm culture dish). After a one day culture, the transfected cells were harvested by trypsinization, washed, and resuspended in 30 ml of Hank's Balanced Salt Solution (HBSS) containing 5 mM HEPES (pH 7.4). The cell suspension was seeded in a 96 well plate (cell plate) at a volume of 90 μl per well and incubated for 30 min in a $CO_2$ incubator. Test compounds, diluted in 0.01% bovine serum albumin (BSA) and HEPES-containing HBSS, were added to the cell plate at a volume of 10 μl per well. After a 1 h incubation in the $CO_2$ incubator, the conditioned media (80 μl) was transferred to an empty 96-well plate (conditioned media (CM) plate). The AP reaction solution (10 mM p-nitrophenylphosphate (p-NPP), 120 mM Tris–HCl (pH 9.5), 40 mM NaCl, and 10 mM $MgCl_2$) was dispensed into the cell plates and the CM plates (80 μl per well). The absorbance at 405 nm ($Abs_{405}$) of the plates was measured, using a microplate reader (SpectraMax 340 PC384, Molecular Devices), before and after a 1-h incubation at room temperature. AP-TGFα release was calculated as described previously[28]. The AP-TGFα release signals were fitted to a four-parameter sigmoidal concentration-response curve, using the Prism 7 software (GraphPad Prism), and the $pEC_{50}$ (equal to $-Log_{10} EC_{50}$) and $E_{max}$ values were obtained.

**β-arrestin recruitment assay**. For the NanoBiT-β-arrestin recruitment assay[54], a receptor construct was designed to fuse the small fragment (SmBiT) of the NanoBiT complementation luciferase to the C-terminus of the ETR construct with a 15-amino acid flexible linker (GGSGGGGSGGSSSGG). A PCR-amplified ETR fragment and an oligonucleotide-synthesized SmBiT were assembled and inserted into the pCAGGS mammalian expression plasmid (a kind gift from Dr. Jun-ichi Miyazaki, Osaka University), using a NEBuilder HiFi DNA Assembly system (New England Biolabs). A β-arrestin construct was generated by fusing the large fragment (LgBiT), with nucleotide sequences gene-synthesized with mammalian codon optimization (Genscript), to the N-terminus of human β-arrestin1 (βarr1) with the 15-amino acid linker. The R393E and R395E mutations, which were shown to abrogate AP-2 binding and thus are defective in internalization[55], were introduced into LgBiT-βarr1 to facilitate the formation of the ETR-βarr1 complex. The plasmid encoding the $ET_B$-SmBiT construct or the $ET_A$-SmBiT construct was transfected, together with the plasmid encoding the internalization-defective

LgBiT-βarr1, into HEK293A cells by the PEI method (1 μg ETR-SmBiT plasmid, 0.5 μg LgBiT-βarr1 plasmid, and 25 μl of 1 mg/ml PEI solution per 10-cm culture dish). After a one-day culture, the transfected cells were harvested with EDTA-containing Dulbecco's phosphate-buffered saline and resuspended in 10 ml of HBSS containing 5 mM HEPES and 0.01% BSA (BSA-HBSS). The cell suspension was seeded in a 96-well white plate at a volume of 80 μl per well and loaded with 20 μl of 50 μM coelenterazine (Carbosynth), diluted in BSA-HBSS. After an incubation at room temperature for 2 h, the background luminescent signals were measured using a luminescent microplate reader (SpectraMax L, Molecular Devices). Test compounds (6×, diluted in BSA-HBSS) were manually added to the cells (20 μl). After ligand addition, the luminescent signals were measured for 15 min at 20-s intervals. The luminescent signal was normalized to the initial count, and the fold-change values over 5–10 min after ligand stimulation were averaged. The fold-change β-arrestin recruitment signals were fitted to a four-parameter sigmoidal concentration–response, and the $pEC_{50}$ and $E_{max}$ values were obtained as described above.

## Data availability

The atomic coordinates and structure factors of the $ET_B$ receptor have been deposited in the Protein Data Bank (PDB) (https://www.rcsb.org/) with accession codes 6IGK (ET3-bound) and 6IGL (IRL1620-bound). The raw X-ray diffraction images are also available at SBGrid Data Bank (https://data.sbgrid.org/) with IDs 611 and 612, respectively. Other data are available from the corresponding authors upon reasonable request.

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

## Acknowledgements

We thank the members of the Nureki lab and the beamline staff at BL32XU of SPring-8 (Sayo, Japan) for technical assistance during data collection. The diffraction experiments were performed at SPring-8 BL32XU (proposal 2016A2527). This work was supported by JSPS KAKENHI grants 16H06294 (O.N.), 17J30010 (W.S.), 30809421 (W.S.), 15H06862 (K.Y.), 17H05000 (T.N.), and 17K08264 (A.I.), and the Core Research for Evolutional Science, PRESTO from the Japan Science and Technology (JST) Technology Program; the Platform for Drug Discovery, Information, and Structural Life Science from the Ministry of Education, Culture, Sports, Science, and Technology of Japan; and the Japan Agency for Medical Research and Development (AMED) grants the PRIME JP17gm5910013 (A.I.) and the LEAP JP17gm0010004 (A.I. and J.A.), and the National Institute of Biomedical Innovation.

## Author contributions

W.S. designed all of the experiments, purified the ET_B receptor in complex with ET-3, collected data, and refined the structures. T.I. expressed, purified, and crystallized the ET_B receptor in complex with ET-3 and IRL1620, collected data, and refined the structures. A.I., F.M.N.K., and J.A. performed and oversaw the cell-based assays. K.Y. and K.H. developed a pipeline for data collection and processing, and assisted with the structure determination and refinement. The manuscript was prepared by W.S., T.I., A.I., K.Y., T.N., and O.N. T.N. and O.N. supervised the research.

## Additional information

**Competing interests:** The authors declare no competing interests.

