## [Peer Review File · Nature Communications]

Reviewers' comments:

Reviewer #1 (Remarks to the Author):

This manuscript is clear, well written and informative. The authors have a very strong track record and the technical aspects and limitations of modifying the ETB receptor for crystallization have already been reported in their previous papers. The pharmacology results are convincing and as expected.

The following are suggestions for further comments:

(1) This group reported the first crystal structure of the human ETB receptor. In 2017 they extended these studies reporting the crystal structure of human ETB receptor bound to bosentan, the first ET antagonist in clinical use and a high affinity ETB-selective antagonist K-8794 was also co-crystallized. Both compounds superimposed well to the receptor structure and the deduced binding was similar. Can the authors comment on the similarities/ differences in the residues interacting with the full agonist ET-3 and partial agonist IRL-1620. It seems by inspection that no residues in the receptor were common to K-8794 binding and IRL-1620 binding. Is this correct and would be expected? Can the authors clarify please.

(2) In their paper in 2016, the authors proposed a model whereby transmembrane helices one, two, six and seven move and envelop the entire ET peptide, to form a lid-like architecture that covers the orthosteric pocket, predicted to form a very stable complex. This provides one structural explanation for the unusual property of ET-1 in causing long lasting responses and with a slow dissociation rate from the receptor. Of key importance is that a similar mechanism is now proposed for ET-3, interacting with the N-terminus which is more variable compared with ET-1 and a key residue is D8. As with ET-1, the N-terminal tail is anchored to TM7 via a disulphide bond between C90 and C358. The authors imply this is a lid, that the bonds between the two Cys residues remain intact and there is no reforming of this bond with residues in ET-1 or ET-3.

Intriguingly IRL-1620 labelled radioactively with [125I] is reported to bind reversibly and human cloned receptors and some other species. (Nambi et al. Species differences in the binding characteristics of [125I]IRL-1620, a potent agonist specific for endothelin-B receptors. *J Pharmacol Exp Ther.* 1994 268:202-7). The D8 residue is present but not the N-terminus – the authors might like to comment. Nambi et al also noted that the time course experiments of association and dissociation indicated that [125I]IRL-1620 binding to dog and human tissues and human ETB clone was rapid and reversible, whereas in rat tissues, the binding was rapid but slowly dissociating suggesting that this might be due to species differences. Can the authors speculate if there are any structural reason for this? Is this due to internalization in rat receptors, which does not occur in human clones or human tissues?

(3) Minor point

The authors have shown in detailed experiments that in their modified receptor (Figure 1) IRL-1620 is a partial agonist compared with ET-1 or ET-3 ie the maximum response is lower with IRL-1620 (although modestly). Other publications show IRL-1620 to be a full agonist and can vary depending on the preparation as can be the case with other receptors.

Reviewer #2 (Remarks to the Author):

This manuscript describes the crystal structure of the endothelin ETB receptor in complex with the natural ligand ET-3, and with the synthetic ETB selective molecule IRL-1620.

The computer modeling work seems of excellent quality. However, it is surprising the work on ET-3 binding to ETB receptor does not lead to an in-depth comparison with ET-1 binding to ETB. ET-3

is considered the likely main natural agonist of ETB receptor and has similar binding affinity to ETB as ET-1, so understanding of the respective binding modes and roles of ET-1 (activating both receptors) and ET-3 (ETB selective) would have been important.

Concerning IRL-1620, the role of ETB receptors is not only of inducing "nitric oxide-mediated vasorelaxation" as presented here. In pathology and perhaps even physiology, ETB receptor also induces vasoconstriction, cell proliferation, in particular of tumor cells, inflammation, increased vascular permeability etc ((Marshall et al., Br J Pharmacol 1999; Clozel et al. BBRC, 1992; Lahav et al., PNAS, 1998..), so this role is complex and probably not protective, and the need for "better ETB agonists" speculative. This may be why in preclinical studies IRL-1620 only induced a transient increase in tumor blood flow (Rajeshkumar et al., 2005, 2007) and lacked efficacy in Phase 2, and it was not studied in patients with hypertension. Big ET-1 injection, which is considered to best reproduce the physiology of ET-1, does not cause vasodilation but only vasoconstriction. And in pathology the vasodilating role of endothelial ETB receptor disappears at the advantage of smooth muscle cell ETB receptors mediating vasoconstriction (Kakoki et al. Circulation, 1999, Cardillo et al, Hypertension, 1999).

Finally it may be that partial agonists may have advantages vis-à-vis full agonists as for example the possibility to avoid interaction with beta-arrestin and thus prevent receptor internalization and desensitization.

Reviewer #3 (Remarks to the Author):

The presented manuscript X-ray structures of human ETB receptor provide mechanistic insight into receptor activation and partial activation by Shihoya and colleagues characterizes the structural basis of the human ETB receptor in complex with natural peptide agonist endothelin-3 and the highly-selective ETB small peptide agonist IRL1620. The class A GPCRs ETA and ETB, the Endothelin receptors, are activated by endothelins 1-3. Both receptors play a major role in blood pressure regulation such as vasoconstriction (ETA) and vasorelaxation (ETB). To understand the structural basis of highly selective agonists like IRL1620 is of utmost importance to develop new selective drugs.

The study is in principle well performed and with several interesting features. However, the structural interpretation and discussion of IRL1620 as partial agonist is very difficult. All changes of IRL1620 and ET-3 in ETB a relatively small and I'm not sure that all conclusions of partial activation are appropriate. I would suggest to revise the figures and show clearer which elements of the receptor are in an active state and which are blocked or inactive. Overall the study is important after a careful revision of the figures and discussion.

Remarks:

- The IRL1620 functions as a partial agonist. Is this on the limit?? How is the limit for a full agonist defined? Is IRL1620 not nearly a full agonist compared to ET-3?

- (page 5 line 106): I think there is a typo in "ETB receptor were 85% (TGf α shedding assay) and...". Emax of IRL1620 is 88% for ETB receptor.

- (page 5 line 112-114): The EC50 value of IRL1620 for ETB-Y5 is not similar to EC50 for the wild-type receptor in the β -arrestin recruitment assay (Fig. 1d). The authors should clarify this issue.

- (page 5, line 116): There is again a typo in "... both assays (84% and 79% in the TGf α shedding assay and the β -arrestin recruitment assay, respectively)". Emax of IRL1620 for ETB-Y5 is 85% for the β -arrestin recruitment assay.

- Did the authors test biochemical activities of ET-3 and IRL1620 for the ETB-Y5-T4L? Is the thermostabilized ETB-Y5 different to ETB-Y5-T4L?

- The citation for the suite ZOO (K.Y., G.U., K.H., M.Y., and K.H., submitted) is not published at this point. The sentence "This system allowed the convenient collection of a large number of datasets and the determination of the highest-resolution agonist-bound GPCR structures." is a bit weird. Is this software only for GPCRs?

- (page 6 line 128 + Supplementary Fig. 1c, d): The Fo – Fc omit maps are very good. But why is the contour level of c, d and e different. I would suggest to show Fo – Fc omit maps at a similar contour level of 2.0 or higher (for d). The N-terminal amide with a succinyl group and the last two amino acids DE seems to be not good resolved in the electron density of IRL1620. The authors could clarify this issue better. Similar problems has the C-terminus in the structure of the ETB-Y5-T4L in complex with ET-1. The last amino acids have a very low or no electron density.

- (Figure 2c): I strongly suggest the choice of a different color as pink for ET-1/ETB structure. I cannot see any differences to the red ET-3/ETB structure.

- (page 6, line 142): Typo: "... superimposed well (Fig. 2b and Supplementary Fig. 2a–d)" is "... superimposed well (Fig. 2c,d and Supplementary Fig. 2a–d)".

- I would suggest a binding plot to see all interactions (hydrophobic and potential hydrogen bonds) of ET-3 and IRL1620 as a supplementary figure.

- For figure 2e I suggest a different coloring for the ligands ET-1 and ET-3 as the receptor color.

- As remarked before the negative cluster of IRL1620 (succinyl group, D8, E9) seems to be not well resolved. The contributions of this N-terminus might be only electrostatic without any specific binding to the receptor. What is the meaning of "the N-terminal dipole moment of the α -helical region" of IRL1620? The authors can discuss this feature in a better way.

- Is there any role of ECL1 in agonist binding in comparison of ETA and ETB? ETA-ECL2 is four amino acids elongated as in ETB.

- Regarding to the active state conformation of TM7 and H8 in ET-3- and IRL1620-bound structures: Is the Tyr of the NPxxY motif moved in the helical bundle? The complete activation (movements of TM5 and 6 and rearrangement of the complete cytoplasmic binding site) is blocked or hindered by T4L, right? On this basis, the conclusion of the extracellular activation might be appropriate but the complete activation path to the cytoplasm is very difficult to interpret. The authors should clarify and discuss this point in the manuscript.

- I cannot see any downward movements of W336 and 147 in figures 6a-c. This figure should be revised to visualize this important feature (Superimpositions, arrows...)

- I would be careful with the interpretation of potential water molecules due to the different resolution of the three structures (especially for the IRL1620-bound structure). The authors could clarify this point in the manuscript.

Following are our item-to-item responses to the specific points raised by the Referees.

Specific comments by Referee #1:

The following are suggestions for further comments:

(1) This group reported the first crystal structure of the human ETB receptor. In 2017 they extended these studies reporting the crystal structure of human ETB receptor bound to bosentan, the first ET antagonist in clinical use and a high affinity ETB-selective antagonist K-8794 was also co-crystallized Both compounds superimposed well to the receptor structure and the deduced binding was similar. Can the authors comment on the similarities/ differences in the residues interacting with the full agonist ET-3 and partial agonist IRL-1620. It seems by inspection that no residues in the receptor were common to K-8794 binding and IRL-1620 binding. is this correct and would be expected? Can the authors clarify please.

We added the sentence “IRL1620 forms essentially similar receptor interactions to those of the α -helical and C-terminal regions of ET-3” (lines 214 to 215) and the description of the minor difference in their binding modes (figure legend of Supplementary Fig. 3). The binding site for the antagonist K8794 overlaps with that for the C-terminal region of IRL1620. To clarify this point, we added the binding plots of ET-3 and IRL1620 (Supplementary Fig. 4a, b).

(2) In their paper in 2016, the authors proposed a model whereby transmembrane helices one, two, six and seven move and envelop the entire ET peptide, to form a lid-like architecture that covers the orthosteric pocket, predicted to form a very stable complex. This provides one structural explanation for the unusual property of ET-1 in causing long lasting responses and with a slow dissociation rate from the receptor. Of key importance is that a similar mechanism is now proposed for ET-3, interacting with the N-terminus which is more variable compared with ET-1 and a key residue is D8. As with ET-1, the N-terminal tail is anchored to TM7 via a disulphide bond between C90 and C358. The authors imply this is a lid, that the bonds between the two Cys residues remain intact and there is no reforming of this bond with residues in ET-1 or ET-3.

Intriguingly IRL-1620 labelled radioactively with [125I] is reported to bind reversibly and human cloned receptors and some other species. (Nambi et al. Species differences in the binding characteristics of [125I]IRL-1620, a potent agonist specific for endothelin-B receptors. J Pharmacol Exp Ther. 1994 268:202-7). The D8 residue is present but not the N-terminus – the authors might like to comment. Nambi et al also noted that the time course experiments of association and dissociation indicated that [125I]IRL-1620 binding to dog and human tissues and human ETB clone was rapid and reversible, whereas in rat tissues, the binding was rapid but slowly dissociating suggesting that this might be due to species differences. Can the

authors speculate if there are any structural reason for this? Is this due to internalization in rat receptors, which does not occur in human clones or human tissues?

We greatly appreciate the Referee for this insightful comment. Due to the lack of the N-terminal region, the α -helical region of IRL1620 seems rather flexible as compared with that of ET-1, which may account for the reversible binding of IRL1620. We added the sentence “Nevertheless, the α -helical region of IRL1620 is less visualized in the electron density, suggesting its higher flexibility as compared to those of ET-1 and ET-3, probably due to the lack of the N-terminal region. In contrast to the quasi-irreversible binding of ET-1, IRL1620 binding is reportedly reversible. Such structural differences may account for the different dissociation properties of these agonists” (lines 181-186). With regard to the slow dissociation kinetics of IRL1620 in rat tissues, the current structure unfortunately does not provide any insights, because the residues constituting the IRL1620 binding site are highly conserved between the human and rat receptors.

(3) Minor point

The authors have shown in detailed experiments that in their modified receptor (Figure 1) IRL-1620 is a partial agonist compared with ET-1 or ET-3 ie the maximum response is lower with IR-1620 (although modestly). Other publications show IRL-1620 to be a full agonist and can vary depending on the preparation as can be the case with other receptors.

In the previous study, the authors evaluated the efficacy of IRL1620 by contraction assays, using isolated guinea pig trachea (Takai, M. *et al. Biochem. Biophys. Res. Commun.* **184**, 953-959 (1992)). In the current study, we re-evaluated the agonist efficacy of IRL1620 by TGF α -shedding and β -arrestin recruitment assays, using HEK 293 cells heterogeneously expressing the human ET_B receptor. These assays allow a more accurate evaluation of the downstream signaling levels, and therefore, we believe that our method is more suitable to evaluate the efficacy of IRL1620, as compared to the “classical” contraction assay.

Specific comments by Referee #2:

The computer modeling work seems of excellent quality. However, it is surprising the work on ET-3 binding to ETB receptor does not lead to an in-depth comparison with ET-1 binding to ETB. ET-3 is considered the likely main natural agonist of ETB receptor and has similar binding affinity to ETB as ET-1, so understanding of the respective binding modes and roles of ET-1 (activating both receptors) and ET-3 (ETB selective) would have been important.

As this Referee pointed out, ET-3, rather than ET-1, has been considered to be the main

natural agonist for ET_B. However, the current ET-3 bound structure revealed only minor differences in their receptor interactions, and this is mentioned in the revised figure legend of Supplementary Fig. 3. Consistently, the ET_B-selectivity of ET-3 is weak in our cell-based assays (Fig. 1b, about 5-fold). These results indicate that ET_B can bind both agonists with similar high affinities, and therefore, we suppose that the different agonist preference of ET_B *in vivo* is probably due to the different expression levels and distributions of the two receptor subtypes (ET_A and ET_B) and the three isopeptides (ET-1 to ET-3). We believe that the current results will facilitate further investigations *in vivo*, which are unfortunately beyond the scope of this study. Nonetheless, we appreciate this comment, as the revised manuscript would provide better information for the readers to understand the slight differences between ET-1 and ET-3.

Concerning IRL-1620, the role of ETB receptors is not only of inducing "nitric oxide-mediated vasorelaxation" as presented here. In pathology and perhaps even physiology, ETB receptor also induces vasoconstriction, cell proliferation, in particular of tumor cells, inflammation, increased vascular permeability etc ((Marshall et al., Br J Pharmacol 1999; Clozel et al. BBRC, 1992; Lahav et al., PNAS, 1998.), so this role is complex and probably not protective, and the need for "better ETB agonists" speculative. This may be why in preclinical studies IRL-1620 only induced a transient increase in tumor blood flow (Rajeshkumar et al., 2005, 2007) and lacked efficacy in Phase 2, and it was not studied in patients with hypertension. Big ET-1 injection, which is considered to best reproduce the physiology of ET-1, does not cause vasodilation but only vasoconstriction. And in pathology the vasodilating role of endothelial ETB receptor disappears at the advantage of smooth muscle cell ETB receptors mediating vasoconstriction (Kakoki et al. Circulation, 1999, Cardillo et al, Hypertension, 1999).

Finally it may be that partial agonists may have advantages vis-à-vis full agonists as for example the possibility to avoid interaction with beta-arrestin and thus prevent receptor internalization and desensitization.

This is a very important point. As this Referee indicated, the ET_B receptor also induces vasoconstriction. Therefore, the two receptor subtypes do not function in a simple “balanced” manner as described in the current manuscript, and this makes it difficult to develop drugs by inducing subtype-specific activation/inhibition. However, our pharmacological and structural studies for the first time revealed the partial agonistic property of IRL1620, which will potentially enable further tuning of its efficacy. Therefore, we revised the discussion, and stated that “The development of ET_B-selective agonists by fine-tuning their G-protein

activation and/or β -arrestin recruitment activities might be beneficial for clinical applications” (lines 333-335).

Specific comments by Referee #3:

- The IRL1620 functions as a partial agonist. Is this on the limit?? How is the limit for a full agonist defined?
Is IRL1620 not nearly a full agonist compared to ET-3?

In this study, we defined ET-1 as the full agonist, as it is the most potent agonist for ET_A and ET_B. Technically, we defined the E_{max} value of a four-parameter sigmoid curve for ET-1 as the full agonist limit (100%). We obtained this value from the “Top” parameter in the “Best-fit values” of the “Variable slope (four parameters)” in the GraphPad Prism 7 software. Likewise, we obtained the E_{max} values of ET-3 and IRL 1620 from the four-parameter sigmoid curves. For each experiment, we normalized the E_{max} values of ET-3 and IRL 1620 to that of ET-1. From n = 4 to 7 experiments, we calculated the means and SEM of the normalized E_{max} values, which are shown in Figures 1b and d. We analyzed whether the normalized E_{max} values (ET-3 and IRL 1620) are significantly different from that of ET-1. In both assays (G-protein activation and β -arrestin recruitment) and both ET_B constructs (ET_B-WT and ET_B-Y5), the E_{max} values of IRL 1620 are significantly smaller than those of ET-1, whereas the E_{max} values of ET-3 are not significantly different from those of ET-1.

The concentration-response curves for ET_B-WT and ET_B-Y5 (Fig. 1a and c) showed that all three of the tested agonists (ET-1, ET-3 and IRL 1620) gave plateau signals (e.g., at least two highest concentrations gave saturated responses). Thus, together with the statistically different E_{max} values of the fitted curves, we conclude that IRL 1620 behaves as a partial agonist for ET_B.

We include a figure in which the β -arrestin recruitment signals were normalized to the E_{max} value of ET-1 for each experiment (Fig. L1). We note that while the mean values are identical (except for the y-axis scales) between Figure 1c and the attached figure, the error bars are smaller in the attached figure because the β -arrestin recruitment signals were normalized in every experiment, which could cancel multiple factors affecting signals (e.g., receptor

expression, cell conditions).

Figure L1 Pharmacological characterizations of ET-3 and IRL1620.

We performed a statistical analysis and confirmed that the E_{\max} values of IRL 1620 are significantly lower than those of ET-3 in both assays (G-protein activation and β -arrestin recruitment) for both ET_B constructs (ET_B -WT and ET_B -Y5). Thus, even if we use ET-3 as a reference full agonist, IRL 1620 can be defined as a partial agonist. We assume that the layout of the figure (Fig. 1b and d) was confusing because the columns (test ligands) were not consistent with the dose-dependent curves for the respective receptors (Fig. 1a and c). Accordingly, in the revised manuscript, we converted the columns and rows of Fig. 1b and d, so that the results for the three receptors (ET_B -WT, ET_B -Y5 and ET_A) are arranged consistently throughout Fig. 1a–d: left panels and columns for ET_B -WT, middle for ET_B -Y5, and right for ET_A .

- (page 5 line 106): I think there is a typo in “*ETB receptor were 85% (TGFA shedding assay) and...*”. *E_{max} of IRL1620 is 88% for ETB receptor.*

We corrected it accordingly (line 103).

- (page 5 line 112-114): *The EC₅₀ value of IRL1620 for ETB-Y5 is not similar to EC₅₀ for the wild-type receptor in the β -arrestin recruitment assay (Fig. 1d). The authors should clarify this issue.*

According to this comment, we clearly described this difference in the revised manuscript, “... while the EC_{50} values of IRL1620 were increased for ET_B -Y5 by about 9- and 6-fold in the G-protein coupling and β -arrestin recruitment assays, respectively, as compared to the wild type receptor” (lines 112-114).

- (page 5, line 116): *There is again a typo in “... both assays (84% and 79% in the TGFA shedding assay and the β -arrestin recruitment assay, respectively)”. E_{max} of IRL1620 for ETB-Y5 is 85% for the β -arrestin recruitment assay.*

We corrected it accordingly (line 111).

- *Did the authors test biochemical activities of ET-3 and IRL1620 for the ETB-Y5-T4L? Is the thermostabilized ETB-Y5 different to ETB-Y5-T4L?*

We cannot measure the biochemical activities of ET-3 and IRL1620 for ET_B -Y5-T4L, because the T4L insertion completely disrupts G-protein binding, but our previous study showed that

the T4L insertion does not affect the ET-1 binding (Shihoya, W. *et al.*, *Nature* **537**, 363-368 (2016)). Therefore, we suppose the ligand-binding properties of ET_B-Y5-T4L to be comparable to those of ET_B-Y5 (Figure L2).

Figure L2 Apparent [¹²⁵I]-labelled ET-1 equilibrium dissociation constants (K_d).

	wt	ET _B R-Y5	ET _B R-Y5-T4L	ET _B R-Y5-mT4L
K _d for [¹²⁵ I]ET-1 (pM)	20.7 ± 1.5	25.3 ± 2.9	26.2 ± 2.7	29.8 ± 2.2

Values of the apparent dissociation constants for the wild-type (WT), thermostabilized (ET_BR-Y5), T4-fused (ET_B-Y4-T4L), and mT4L-fused (ET_B-Y5-mT4L) constructs are shown.

- The citation for the suite ZOO (K.Y., G.U., K.H., M.Y., and K.H., submitted) is not published at this point. The sentence “This system allowed the convenient collection of a large number of datasets and the determination of the highest-resolution agonist-bound GPCR structures.” is a bit weird. Is this software only for GPCRs?

As pointed out by Referee #3, the automated data collection system ZOO can be applied for the data collection of all types of protein microcrystals. We rephrased this as “which allowed the convenient collection of a large number of datasets and the determination” (lines 123-124).

- (page 6 line 128 + Supplementary Fig. 1c, d): The F_o – F_c omit maps are very good. But why is the contour level of c, d and e different. I would suggest to show F_o – F_c omit maps at a similar contour level of 2.0 or higher (for d). The N-terminal amide with a succinyl group and the last two amino acids DE seems to be not good resolved in the electron density of IRL1620. The authors could clarify this issue better. Similar problems has the C-terminus in the structure of the ET_B-Y5-T4L in complex with ET-1. The last amino acids have a very low or no electron density.

In the original manuscript, we showed the 2F_o – F_c maps for ET-3 and IRL1620, by mistake. We replaced these maps with the F_o – F_c omit maps contoured at 2.0σ (Supplementary Fig. 1c, d). In the map of IRL1620, D8, E9, and the N-terminal succinyl group do not have well resolved electron densities, as pointed out by Referee #3. We also added the description in the figure legend of Supplementary Fig. 1c, d.

- (Figure 2c): I strongly suggest the choice of a different color as pink for ET-1/ET_B structure. I cannot see any differences to the red ET-3/ET_B structure.

According to the suggestion, the ET-1/ET_B structure was coloured green (Fig. 2c).

- (page 6, line 142): Typo: "... superimposed well (Fig. 2b and Supplementary Fig. 2a-d)" is "... superimposed well (Fig. 2c,d and Supplementary Fig. 2a-d)".

We corrected it accordingly (line 139).

- I would suggest a binding plot to see all interactions (hydrophobic and potential hydrogen bonds) of ET-3 and IRL1620 as a supplementary figure.

According to the suggestion, we added the binding plots of ET-3 and IRL1620 (Supplementary Fig. 4a, b).

- For figure 2e I suggest a different coloring for the ligands ET-1 and ET-3 as the receptor color.

According to the suggestion, ET-1 and ET-3 were coloured differently: N-terminal region in cyan, α -helical region in orange, and C-terminal region in deep pink).

- As remarked before the negative cluster of IRL1620 (succinyl group, D8, E9) seems to be not well resolved. The contributions of this N-terminus might be only electrostatic without any specific binding to the receptor. What is the meaning of "the N-terminal dipole moment of the α -helical region" of IRL1620? The authors can discuss this feature in a better way.

In general, negatively-charged residues at the N-terminal end of α -helices stabilize the helical conformation, by forming hydrogen bonds with the exposed amines of the helix (Int J Pept Protein Res. 41, p499-511 and refX in the manuscript). In the current structure, IRL1620 has a negative charge cluster at the N-terminal end of the α -helical region, which probably stabilizes its helical conformation and thus facilitates the overall interaction with the receptor, as well as the docking to the receptor in an electrostatic complementary manner. These points are clarified in the revised manuscript. "Moreover, this negative cluster probably stabilizes the α -helical conformation of IRL1620, by forming a hydrogen-bonding cap at the N-terminally-exposed amines, which may facilitate the overall receptor interactions" (lines 174-177). As noted by Referee #3, the electron density of the terminal residues is less visible, which might be due to the higher flexibility of the α -helical region, as compared to those of ET-1 and ET-3. These features are discussed in the revised manuscript (lines 181-186).

- Is there any role of ECL1 in agonist binding in comparison of ETA and ETB? ETA-ECL2 is four amino acids

elongated as in ETB.

According to the suggestion, we added the discussion about the effect of the elongation of ET_A-ECL1 (lines 203-209 and Fig. 4a).

Regarding to the active state conformation of TM7 and H8 in ET-3- and IRL1620-bound structures: Is the Tyr of the NPxxY motif moved in the helical bundle?

Y7.53 in the NPXXY motif is replaced with leucine in both the ET_A and ET_B receptors, (Supplementary Fig. 5a), but the agonist-induced inward motion of TM7 is somewhat conserved, as discussed in detail in the previous study (Shihoya, W. *et al.*, *Nature* **537**, 363-368 (2016)).

The complete activation (movements of TM5 and 6 and rearrangement of the complete cytoplasmic binding site) is blocked or hindered by T4L, right? On this basis, the conclusion of the extracellular activation might be appropriate but the complete activation path to the cytoplasm is very difficult to interpret. The authors should clarify and discuss this point in the manuscript.

We agree with Referee #3 that the intracellular movements of TMs 5 and 6 are hindered by the T4L-fusion. However, ET-3 binding induced a 1 Å outward displacement of the middle part of TM6. In contrast, IRL1620 binding does not induce this displacement, revealing that IRL1620 actually induces different conformational changes in the receptor core, which probably affect the dynamics of the cytoplasmic side of TM6. To clearly describe this difference, we modified Fig. 5d–f to show the movement of the middle part of TM6 and added the phrase “the middle part of TM6” (line 247).

- I cannot see any downward movements of W336 and 147 in figures 6a-c. This figure should be revised to visualize this important feature (Superimpositions, arrows...)

According to the suggestion, we added arrows to clarify the downward movements (Fig. 6).

- I would be careful with the interpretation of potential water molecules due to the different resolution of the three structures (especially for the IRL1620-bound structure). The authors could clarify this point in the manuscript.

First, we should note that even though the resolution of the IRL1620-bound structure is lower than other high-resolution structures (*e.g.*, ET-3-bound structure), it revealed an apparently different conformation, including the smaller tilting of TM6 and the water displacement in the

receptor core. However, we agree that we should be careful with the interpretation of the water molecules in the IRL1620 bound structure. We suppose that not only the water molecule displacement, but different tilting of TM6 itself also affects G-protein activation level. Accordingly, we revised as follows, “Despite the lower resolution of the IRL1620-bound structure, the $2F_o - F_c$ map shows different rearrangement of water molecules and amino acid residues in the receptor core, in which the hydrogen-bonding network is partially preserved (Fig. 5e, Supplementary Fig. 7c). This preserved network, together with the smaller inward motion of TM6, may prevent cytoplasmic outward motion of TM6 that occurs upon G-protein binding.” (lines 288-293).

With these modifications, we hope that our revised manuscript is now suitable for publication in *Nature Communications*.

REVIEWERS' COMMENTS:

Reviewer #3 (Remarks to the Author):

The revised manuscript "X-ray structures of human ETB receptor provide mechanistic insight into receptor activation and partial activation" by Shihoya and colleagues is significantly improved. The revised manuscript had addressed all my issues. Overall, I would recommend the publication. There are only very minor (but several) typos and issues in the manuscript.

For example:

- line 134: Refers to Fig. 2b instead of Fig. 2a,b
- For Pro90 is no van der Waals contact visible. The authors could correct the sentence (line 612) to something like "... two of them form van der Waals interactions". Note that the authors wrote in the main text (line 158-159): "In addition, three consecutive prolines (P87, 88, 89)..." Is here a mix-up?
- There are some typos in the manuscript such as line 614 "van der Waals" instead of "van del Waals"
- I suggest to shift and combine figure S12 with figure 2 (or insert as additional new figure 3) and label amino acids in ET-1, ET-3, and IRL1620 but not in the superimposed figure. It is a very good view and a fantastic orientation for the reader about the similarities and differences of all three ligands.
- The authors could add citations to the sentence line 434-436): " This is consistent with the notion that the fully-active conformation is only stabilized when the G-protein is bound, as shown in the previous nuclear magnetic resonance (NMR) and double electron electron resonance (DEER) spectroscopy studies of GPCRs."

REVIEWERS' COMMENTS:

Reviewer #3 (Remarks to the Author):

The revised manuscript “X-ray structures of human ETB receptor provide mechanistic insight into receptor activation and partial activation” by Shihoya and colleagues is significantly improved. The revised manuscript had addressed all my issues. Overall, I would recommend the publication.

We appreciate reviewer#3’s fruitful suggestions and favorable comments on our manuscript.

There are only very minor (but several) typos and issues in the manuscript.

For example:

- line 134: Refers to Fig. 2b instead of Fig. 2a,b

We corrected accordingly.

- For Pro90 is no van der Waals contact visible. The authors could correct the sentence (line 612) to something like “... two of them form van der Waals interactions”. Note that the authors wrote in the main text (line 158-159): “In addition, three consecutive prolines (P87, 88, 89)...” Is here a mix-up?

- There are some typos in the manuscript such as line 614 “van der Waals” instead of “van del Waals”

We corrected accordingly (line 177-178).

- I suggest to shift and combine figure SI2 with figure 2 (or insert as additional new figure 3) and label amino acids in ET-1, ET-3, and IRL1620 but not in the superimposed figure. It is a very good view and a fantastic orientation for the reader about the similarities and differences of all three ligands.

We appreciate for this comment. Given the integrity of the order of the figures, we would prefer as in the present form.

- The authors could add *citations* to the sentence line 434-436):” This is consistent with the notion that

the fully-active conformation is only stabilized when the G-protein is bound, as shown in the previous nuclear magnetic resonance (NMR) and double electron electron resonance (DEER) spectroscopy studies of GPCRs.”

We added the citation for these descriptions.